# Probing the Structural Determinants of Amino Acid Recognition: X-Ray Studies of Crystalline Ditopic Host-Guest Complexes of the Positively Charged Amino Acids, Arg, Lys, and His with a Cavitand Molecule

**DOI:** 10.3390/molecules23123368

**Published:** 2018-12-19

**Authors:** Giovanna Brancatelli, Enrico Dalcanale, Roberta Pinalli, Silvano Geremia

**Affiliations:** 1Centre of Excellence in Biocrystallography, Department of Chemical and Pharmaceutical Sciences, University of Trieste, 34127 Trieste, Italy; giovanna_brancatelli@hotmail.com; 2Department of Chemistry, Life Sciences and Environmental Sustainability, University of Parma, and INSTM, UdR Parma, Parco Area delle Scienze 17/A, 43124 Parma, Italy; enrico.dalcanale@unipr.it

**Keywords:** amino acid, phosphonate cavitand, molecular recognition, X-ray structure, host-guest chemistry, lysine, arginine, histidine

## Abstract

Crystallization of tetraphosphonate cavitand Tiiii[H, CH_3_, CH_3_] in the presence of positively charged amino acids, namely arginine, lysine, or histidine, afforded host-guest complex structures. The X-ray structure determination revealed that in all three structures, the fully protonated form of the amino acid is ditopically complexed by two tetraphosphonate cavitand molecules. Guanidinium, ammonium, and imidazolium cationic groups of the amino acid side chain are hosted in the cavity of a phosphonate receptor, and are held in place by specific hydrogen bonding interactions with the P=O groups of the cavitand molecule. In all three structures, the positively charged α-ammonium groups form H-bonds with the P=O groups, and with a water molecule hosted in the cavity of a second tetraphosphonate molecule. Furthermore, water-assisted dimerization was observed for the cavitand/histidine ditopic complex. In this 4:2 supramolecular complex, a bridged water molecule is held by two carboxylic acid groups of the dimerized amino acid. The structural information obtained on the geometrical constrains necessary for the possible encapsulation of the amino acids are important for the rational design of devices for analytical and medical applications.

## 1. Introduction

l-Arginine (Arg), l-lysine (Lys), and l-histidine (His) belong to the class of 20 proteinogenic amino acids. Arg is classified as a conditionally essential amino acid for adults [1], and it serves as a precursor for methylguanidoacetic acid (creatine), which plays an essential role in the energy metabolism of muscle, nerve, and testis, and accounts for Arg catabolism of (4-aminobutyl)guanidine (agmatine) by decarboxylation. Arg is involved as a substrate in the urea cycle, and in nitric oxide production, the removal of ammonia from the body, cell division, and functioning of the immune system [2]. Lysine is an essential amino acid that plays an important role in the solvation, structure, and activity of proteins [3]. Arg and Lys are mostly exposed to the globular protein surface, and they are involved in the electrostatic interactions in protein-protein and protein-DNA complexes. Furthermore, through salt bridge formation with the complementary Glu and Asp residues, they contribute to the protein stability [4,5]. Arg and Lys side chain methylations are fundamental in post-translational modifications of histones, essential in epigenetic regulation. Methylations occur primarily at Lys and Arg residues on the histone backbone [6], affecting transcriptional regulation [7], assembly of heterochromatin, and cell cycle progression [8].

His is a conditionally essential amino acid, and plays an important role in the growth and repair of tissues, in the control of the metal elements transmission in biological bases [9], and in protein interactions [10]. Histidine is present in the naturally occurring histidine-rich protein II (HRP2), a common target for rapid diagnostic tests for malaria [11]

The unique structures of the three proteinogenic basic amino acids, Arg, Lys, and His, having long and positively charged side chains, make them interesting targets among the twenty amino acids for the molecular recognition of a specific side chain in a peptide [10,12].

In the literature, there are many examples reporting the use of synthetic macrocycles for Arg and Lys amino acid recognition [12,13,14,15], while the use of molecular receptors for histidine recognition is less explored [16].

Cavitands are a class of abiotic macromolecular receptors based on a resorcinarene scaffold having enforced cavities of molecular dimensions [17,18]. In the design of cavitands, the choice of the bridging groups connecting the phenolic hydroxyls of the resorcinarene scaffold determines shape, rigidity, dimensions, and complexation properties of the resulting cavity [19]. The selection can be made based on the particular class of analytes to be detected. Besides shape complementary, the selective recognition of a guest by a cavitand host requires the presence of specific weak interactions such as hydrogen bonding [20], π-π stacking [21], CH-π [22], and cation-π interactions [23]. The degree of sophistication achieved in controlling weak host–guest interactions in cavitands is such that it allows the rational design of synthetic receptors according to the analyte to be detected. The different functionalization of the cavity upper rim leads to the synthesis of cavitands presenting remarkable molecular recognition properties towards different guests, like aromatic and halogenated hydrocarbons [24,25,26,27,28], short chain alcohols [29,30], and *N*-methylammonium salts [31]. The ability of the cavitands to selectively recognize analytes can be exploited in solution [31], and at the gas-solid [25,26] and solid-liquid interface [28,30]. In particular, tetraphosphonate cavitands, named Tiiii [32], were successfully employed in the molecular recognition of biological relevant *N*-methyl ammonium salts, like sarcosine in urine [33,34]. The origin of Tiiii selectivity toward these species can be attributed to the presence of three different interaction modes: (i) N^+^···O=P cation-dipole interactions; (ii) cation-π interactions of the ^+^N–CH_3_ group with the π basic cavity; and (iii) two simultaneous hydrogen bonds between two adjacent P=O bridges and the two nitrogen protons. Recently, the ability of Tiiii in selectively complex different amino acids, both in solution and in the solid state, was demonstrated [35]. The work focused on the interaction of Tiiii with the ammonium/*N*-methyl ammonium group of 13 amino acids, neglecting the complexation of amino acids with biologically relevant substituents like guanidinium (Arg) and imidazole (His). These side chains are relevant targets for protein camouflage, since molecular receptors that recognize protein surfaces are important tools for recognition and activity modulation of proteins [36,37,38,39,40]. Sulfonatocalix[4]arene and cucurbiturils turned out to be the preferred synthetic receptors for protein surface recognition [41,42,43].

A recent work of Paton and co-workers describes the cation-π interactions of neutral aromatic ligands, using benzene as an archetype, with the cationic amino acid residues Arg and Lys via ab initio calculations, symmetry-adapted perturbation theory (SAPT), and a systematic meta-analysis of all available Protein Data Bank (PDB) X-ray structures [44]. The Lys-arene interaction is predicted to be weakened by polar surroundings to the point that it has a negligible effect on an aromatic ligand binding mode. By contrast, the cation-π interaction made by Arg residues with aromatic ligands is more robust to changes in the surrounding environment.

Here, we report the crystal structures of three complexes between the Tiiii cavitand as host and Arg, Lys, and His as guests. The purpose of the study is to determine which are the interactions responsible for amino acid recognition in the three cases, and to identify the role of the amino acids side chain in the complexation event.

## 2. Results and Discussion

Within a systematic crystallization program to assess the complexation properties of tetraphosphonate cavitands towards amino acids in the solid state [36], we obtained crystals of the fully protonated form of proteinogenic positively charged amino acids, such as Arg, Lys, and His. Crystallization trials were performed using the vapor diffusion method with sitting drops technique in Linbro multi-well plates containing trifluoroethanol (TFE) as solvent and PEG300 as precipitant (see Material and Methods). Within the class of tetraphosphonate cavitand, Tiiii[H, CH_3_, CH_3_], a compact synthetic receptor without substituents at its lower rim (Scheme 1), was chosen for its great tendency to crystallize. Then, accurate crystal structures of the Arg, Lys, and His amino acids complexed with the tetraphosphonate cavitand were determined by single crystal X-ray diffraction using synchrotron radiation as X-ray source, with crystals quenched at cryogenic temperatures (see Material and Methods).

The analysis of the diffraction patterns revealed that the single crystals obtained by co-crystallization of the cavitand in the presence of the fully protonated amino acids were monoclinic (P2_1_ space group) for the Arg and His complexes, while the Tiiii/Lys complex crystallized in the orthorhombic P2_1_2_1_2 space group. The asymmetric unit of the 2 Tiiii[H, CH_3_, CH_3_]•Arg·2HCl and 2 Tiiii[H, CH_3_, CH_3_]•Lys·2HCl crystal structures is composed of two host and one guest molecule, forming a 2:1 host-guest ditopic complex (Figure 1a,b), while, in the 2 Tiiii[H, CH_3_, CH_3_]•His·2HCl crystal structure, four crystallographically independent cavitands and two amino acids molecules form a sort of dimeric 2:1 host-guest ditopic complex (Figure 1c).

In all three structures, a chloride ion was modeled at the lower rim of the cavitand, forming weak C–H···Cl^−^ interactions with the aromatic CH fragments (green spheres in Figure 1).

The relevant geometric parameters describing the host-guest interactions are reported in Table 1. The X-ray structure determination revealed two binding geometries for the side chain fragments of arginine, lysine, and histidine. Guanidinium, ammonium, and imidazolium cationic groups of the side chain are hosted in the cavity of the phosphonate receptor, and are held in place by specific hydrogen bonding interactions with the P=O groups of a cavitand molecule (Figure 2 and Figure 3).

The guanidinium group of Arg was found to be statistically disordered over two equally populated positions (conformation I and II). In conformation I, the side chain of Arg interacts with all four P=O groups of Tiiii (Table 1 and Figure 2a,b), and its terminal atoms are inserted below the main plane, defined by the oxygen atoms of the P=O groups. The planar guanidinium group forms a dihedral angle of 40° with the mean plane of the four oxygen atoms of the P=O groups of Tiiii host. In conformation II, the planar guanidinium group is more perpendicular with respect to the oxygen atoms’ plane (dihedral angle of 63°), and forms only two strong H-bonds with two opposite P=O groups (Table 1 and Figure 2c,d). In this case, a NH_2_ group is located in the center of the cavity of the tetraphosphonate host, and it is slightly more deeply inserted in the cavity with respect to the other conformation (Table 1 and Figure 2b,d).

The ammonium group of Lys, located above the P=O plane, interacts and forms strong H-bonds with two adjacent P=O groups of Tiiii (Table 1 and Figure 3a,b) and with a water molecule inserted in the Tiiii cavity. This internal water molecule forms two H-bonds with the remaining oxygen atoms of the P=O groups (Table 1 and Figure 3a,b).

In the His crystal, both crystallographically independent 2:1 host-guest complexes have a very similar side chain interaction with the cavitand (Figure 1c). The imidazolium cationic group of the His amino acid is inserted with its C_ε1_ atom into the center of the cavity, and it forms two strong H-bonds with two opposite P=O groups (Table 1 and Figure 3c,d). The dihedral angle between the imidazolium group and the oxygen atoms plane of Tiiii is 57° and 67°, in the two crystallographically independent complexes, respectively. It is interesting to note that, while the interaction of the Lys side chain is assisted by a buried water molecule, no extra water molecules are presented in the cavity of Tiiii hosting the Arg or His side chains. This is probably due to the fact that the guanidinium and imidazolium groups are able to form H-bonds with two opposite P=O groups (see Figure 2c and Figure 3c). In such arrangements, a water molecule does not have enough space inside the cavity and any possibility to form double H-bonds with two vicinal P=O groups, as in the case of the complexes with the ammonium group (Figure 3a and Figure 4a). On the other hand, Lys side chain has a higher charge density compared with the Arg and His side chains, in which the conjugation between double bond and nitrogen lone pairs delocalizes the positive charge. Consequently, desolvation of the planar guanidinium and imidazolium groups should be more favorable with respect to the ammonium group of Lys amino acid.

In all three structures, the fully protonated form of the amino acid is ditopically complexed (Figure 1). The positively charged α-ammonium groups form H-bonds with the P=O groups, and with a water molecule hosted in the cavity of a second tetraphosphonate molecule. The nitrogen atom is above the O mean plane, defined by the oxygen atoms of the P=O units, while the water molecule is located below this plane (Figure 4). More specifically, in the Lys complex, this water molecule is much more inserted in the cavity with respect to the analogous water molecule on the opposing cavitand that hosts the ζ-ammonium group (1.15 Å, see Table 1). This different behavior can be associated with the presence, in the α ammonium group complexation, of a methylene β carbon that covers the cavity (Figure 4a). On the other hand, the complexation of the linear side chain of Lys leaves the water molecule free to adopt a different position in the hydrophobic cavity (Figure 3a).

This amino acid/cavitand interaction mediated by a water molecule has been observed in all three complexes, namely with Arg, Lys, and His, and it involves the constant part of the α-amino acid. This “nonspecific” amino acid H-bond interaction was already reported in previously determined host-guest complexes involving the hydrophobic amino acids: Ala, Val, Leu, and Ile [35].

A particular H-bond interaction between the carboxylic acid group of Arg and a P=O group of cavitand mediated by a water molecule has been observed (Figure 5a). This interaction, noted only in 2 Tiiii[H, CH_3_, CH_3_]•Arg 2HCl complex, links the two cavitands of the 2:1 complex by an H-bond network that involves the α-ammonium and α-carboxylic acid groups of the guest.

An interesting difference within the positively charge proteinogenic amino acids is the formation, in the solid state, of water-assisted dimerization of the cavitand/histidine 2:1 complex. In this structure, a bridged water molecule is held by two carboxylic acid groups of the dimerized amino acid (Figure 5b). From the topological point of view, this dimer shows that the couple of 2:1 complexes are mutually oriented in such a way that the side chain of the guest amino acid has the same direction and orientation (Figure 1c).

The cavitand units of the 2:1 complexes are in a typical cone conformation with the amino acid guest, connecting the two bowls, oriented slantwise to allow for optimal host-guest interaction. Despite the formation of 2:1 complexes with similar α-ammonium/Tiiii interaction and analogue side chain complexation, the crystal structures of the three amino acid/cavitand complexes show a quite different spatial arrangement of the two cavitands.

The overall shape of the 2 Tiiii[H, CH_3_, CH_3_]•Arg 2HCl complex is quasi-capsular, with the Arg amino acid captured between the two almost-aligned cavitands. In particular, the two bowls, facing each other, hold the Arg amino acid in a clamp-like arrangement. The dihedral angle between the mean planes of the four oxygen atoms of the P=O groups of two cavitands, representing the opening angle of the clamp, is 36° (Figure 6a).

In the 2 Tiiii[H, CH_3_, CH_3_]•Lys 2HCl complex, the two cavitand bowls are almost perfectly anti-parallel, while they are offset by about a half cavitand (Figure 6b). This misalignment significantly opens the clamp formed by the two host molecules.

In the 2 Tiiii[H, CH_3_, CH_3_]•His 2HCl complex, the two cavitand bowls are both shifted laterally, by more than half cavitand, and rotated by about 48° (dihedral angle between the P=O planes), resulting in a more open clamp with respect to the other two cases (Figure 6c).

With respect to the ditopic behavior of positively charge amino acids, an unexpected result obtained in this crystal structure analysis is the trend observed in the opening of the clamp of the 2:1 complex. The trend observed, His > Lys > Arg, is in reverse order with respect to the length of amino acid side chain Arg > Lys > His. This reverse trend can be attributed to dimer formation, in the case of the 2 Tiiii[H, CH_3_, CH_3_]•His 2HCl complex (Figure 1c), and to the different behavior between Arg and Lys residues in the insertion of the side chain into the cavitand (Table 1 and Figure 2b and Figure 3b).

## 3. Materials and Methods

### 3.1. Co-Crystallization Experiments

For the co-crystallization experiments, the phosphonate cavitand Tiiii[H, CH_3_, CH_3_], with the P=O groups pointing inward the cavity, was synthesized as previously described [45], and 2,2,2-trifluoroethanol (TFE) and amino acids (l-arginine, l-lysine, and l-histidine) were purchased from Sigma-Aldrich (St. Louis, MO, USA), and used as supplied.

Crystals of host-guest complexes containing amino acids were obtained by co-crystallization microscale experiments with the sitting drop vapor diffusion technique. The co-crystallization solution was prepared by adding, to a TFE solution (30 mM) of cavitand Tiiii[H, CH_3_, CH_3_], a solution containing the desired amino acid. The solutions of amino acids were prepared in 1 M HCl (see Appendix A for experimental details). As a general procedure, 4 μL of the solution containing the cavitand and the amino acid were set on a microbridge, and then 4 μL of the reservoir solution were added to the drop. The drop was left to equilibrate against 1 mL of reservoir solution, containing the organic polymer polyethylene glycol (PEG) 300 in the range between 5 and 50% (*v/v*) as precipitant agent. Crystals were left to grow at the temperatures of 4 °C. After 1 month, good quality crystals suitable for X-ray diffraction analysis were obtained.

### 3.2. Crystal Structure Determination

Data collections were carried out at the Macromolecular crystallography XRD1 beamline of the Elettra synchrotron (Trieste, Italy), by employing the rotating crystal method and the cryo-cooling technique. Routinely, the crystal was mounted in a loop and flash frozen at 100 K with liquid nitrogen without adding further cryoprotectant, thanks to the presence of PEG 300 in the mother liquor. Diffraction data of Arg, Lys, and His/cavitand complexes were indexed and integrated using the XDS package [46]. Scaling was carried out with AIMLESS, for the dataset collected from crystals of Arg, Lys, and His/cavitand supramolecular complexes.

All structures were solved by direct methods using SIR2011 [47]. Non-hydrogen atoms at full occupancy, or with population higher than 0.5, were anisotropically refined (H atoms at the calculated positions) by full-matrix least-squares methods on *F*^2^ using SHELXL-13. Restraints on the geometrical and thermal parameters of the disordered solvent molecules (DFIX, DELU, ISOR) were introduced during the last refinement cycles. Several TFE co-crystallized solvent molecules were found in the asymmetric units for all the structures. In all the structures, electronic density that could be attributed to highly disordered solvent molecules was detected. The contribute of the disordered solvent to the overall scattering was removed through the SQUEEZE function of PLATON software [48]. A detailed discussion of the refinement for each structure is provided below.

#### 3.2.1. Structure Refinement of 2 Tiiii[H, CH_3_, CH_3_]•Arg 2HCl Complex

In the asymmetric unit of the complex of Tiiii[H, CH_3_, CH_3_] with doubly protonated Arg, a dimeric supramolecular host-guest complex was found together with 2.9 TFE solvent molecules and 3.7 water molecules. The guanidinium fragment of arginine was found disordered over two positions refined with equal occupancy. The residual electron density of 155 electrons/cell, found in the voids of the crystal (corresponding to 9% of the cell volume), was attributed to about 1.5 highly disordered CF_3_CH_2_OH molecules in asymmetric units. A refinement using reflections modified by the SQUEEZE procedure [48] behaved well, and the *R*-factor was reduced from 9.1% to 7.5%.

#### 3.2.2. Structure Refinement of 2 Tiiii[H, CH_3_, CH_3_]•Lys 2HCl Complex

In the asymmetric unit of the crystal of the complex of Tiiii[H, CH_3_, CH_3_] with doubly protonated Lys, a dimeric host-guest complex was detected together with 0.8 TFE solvent molecules and 2.8 water molecules. The residual electron density of 1097 electrons/cell, found in the voids of the crystal (corresponding to 28% of the cell volume), was attributed to a disordered chlorine ion and about 5.3 CF_3_CH_2_OH molecules in asymmetric units. A refinement using reflections modified by the SQUEEZE procedure [48] behaved well, and the *R*-factor was reduced from 14.3% to 6.8%.

#### 3.2.3. Structure Refinement of 2 Tiiii[H, CH_3_, CH_3_]•His 2HCl Complex

In the asymmetric unit of the crystal of the complex 2 Tiiii[H, CH_3_, CH_3_] with doubly protonated His, two crystallographically independent dimeric host-guest complexes were detected together with 7.82 TFE solvent molecules and four water molecules. The residual electron density of 191.7 electrons/cell, found in the voids of the crystal (corresponding to 7% of the cell volume), was attributed to about 2.1 CF_3_CH_2_OH molecules in asymmetric units. A refinement using reflections modified by the SQUEEZE procedure behaved well, and the *R*-factor was reduced from 7.4% to 6.8%.

Crystal data and refinement details are reported in Appendix A. CCDC 1869186 (Tiiii/His), 1869187 (Tiiii/Arg), and 1876524 (Tiiii/Lys) contain the supplementary crystallographic data for this paper. These data can be obtained free of charge via http://www.ccdc.cam.ac.uk/conts/retrieving.html (or from the CCDC, 12 Union Road, Cambridge CB2 1EZ, UK; Fax: +44-1223-336033; E-mail: deposit@ccdc.cam.ac.uk)

## 4. Conclusions

Molecular recognition of amino acids is a topic of particular interest because it involves several important applicative aspects, such as advanced therapeutic approaches towards target proteins, as well as sensor applications for biochemical analysis and immobilization techniques for protein purification and/or characterization [38,49]. Within a systematic crystallization program, to assess the complexation properties of a cavitand molecule towards amino acids [35], we obtained crystals of the fully protonated form of the three proteinogenic positively charged amino acids, Arg, Lys, and His complexed with the tetraphosphonate cavitand Tiiii[H, CH_3_, CH_3_] in cone conformation. The X-ray structure revealed that, in all three complexes, the fully protonated form of these amino acids has a ditopic behavior. Each amino acid is complexed by two tetraphosphonate cavitand molecules. The side chain of the amino acid is hosted by a cavitand molecule from one side, while the α-ammonium group interacts with a second cavitand molecule on the opposite side. Then, an overall 2:1 host-guest complex is assembled. In particular, the guanidinium, ammonium, and imidazolium cationic groups of the side chain of the amino acids, Arg, Lys, and His, are respectively hosted in the cavity of a phosphonate receptor, and they are held in place by specific hydrogen bonding interactions with the P=O groups of the cavitand molecule. The positively charged α-ammonium group of the three amino acids forms H-bonds with two P=O groups, and with a water molecule hosted in the cavity of a second tetraphosphonate molecule. The chloride counter ion has a crucial role in the stabilization of this complex because it has been detected at the bottom of the lower rim of both cavitands, just on the opposite site of the cavity. The chloride ions form weak C–H···Cl^−^ interactions with the aromatic CH fragments of both Tiiii units, balancing the two positive charges of the guest. It is interesting to note that, in the solid state, the cavitand/histidine 2:1 complex forms a dimer through the carboxylic acid group of the amino acid. This dimerization, with formation of a supramolecular 4:2 host-guest complex, is assisted by a crucial bridged water molecule. This phenomenon has been observed only for this specific amino acid complex. The analysis of the relative disposition of the cavitands in the 2:1 host-guest complexes has evidenced an unexpected ditopic behavior of the proteinogenic positively charged amino acids. In particular, the trend observed in the opening of the clamp formed by the two cavitands, His > Lys > Arg, is inverted with respect to the trend of the length of amino acid side chain Arg > Lys > His. The bulkiest guest, Arg, is almost completely encapsulated by two phosphonate cavitands, while the shorter Lys and the even shorter His gradually show a more open structure of the complex. The overall architecture observed in these complexes provide support for the rational design of ditopic cavitands as molecular clamps for encapsulation of proteinogenic positively charged amino acids. [50,51].

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
