# Peer review of "Probing the Structural Determinants of Amino Acid Recognition: X-Ray Studies of Crystalline Ditopic Host-Guest Complexes of the Positively Charged Amino Acids, Arg, Lys, and His with a Cavitand Molecule"

_molecules, 2018, doi:10.3390/molecules23123368_

Round 1

Reviewer 1 Report

Brancatelli et al report an interesting X-ray crystallographic study of amino acid recognition by a phosphonate cavitand. The motivation for this type of work is high considering the current interest in supramolecular chemistry with proteins. Here, new insights are provided on the complexation of cationic side chains. The X-ray structures appear to be of high quality and the data interpretation is sound.

The different binding modes between the phosphonate cavitand and the guanidinium, ammonium or imidazolium groups is very nicely illustrated in Figures 1-5. The alternate conformations of the guanidinium, involving 2 or 4 hydrogen bonds, is consistent with the diverse interactions Arg makes in protein structures. Lysine appears to be a special case in that its ammonium group is solvated by one water molecule, which in turn is H-bonded to the cavitand. I suggest a slightly different interpretation of this result, which points to the high charge density on this group compared with the guanidinium and imidazolium cations. Consequently, desolvation of the ammonium group is more unfavourable. Note also the tetrahedral geometry around the ammonium group, completed in this case by the water molecule. The planar guanidinium and imidazolium are less suited to solvation.

I also found the complexation of the alpha ammonium group interesting. Again, a water molecule solvates the amine. But this time the water is “buried” relative to the plane of the cavitand oxygens. A short comment on this detail would be helpful.

Considering the dominance of the P=O group in cation binding - is the charge density on the oxygen known? This information might aid in understanding the strength of the hydrogen bonds with the cationic side chains.

Finally, the introduction could be improved. The opening focus on general biology of Arg, Lys and His is unhelpful. The comment on Tat and Rev is incorrect and some of the cited literature is not the best. It is suggested to focus instead on protein recognition by macrocycles, emphasizing the details of cation binding. For example, cite also Chem. Commun. 2014, 50, 10412, which treats arginine recognition by sulfonato-calixarene.

Minor comments:

It would benefit the community to deposit these structures in the CCDC.

Line 121, replace “In all three structures the chloride anions have been detected at the bottom of the lower rim” with “In all three structures a chloride ion was modelled at the lower rim”.

Line 161, “deeply inserted” in relation to histidine is an over-statement. Deeply buried might suggest involvement of the phenyl rings of the cavitand.

Define the atom colour scheme once only in Figure 1. It is not necessary to repeat in the other figures.

Is Figure 6 really necessary? It appears to be partly a duplication of Figure 1. If retained, use the consistent colour scheme.

Are references 14-16 necessary? One would suffice.

Author Response

Response to Reviewer 1 Comments

Point 1 The different binding modes between the phosphonate cavitand and the guanidinium, ammonium or imidazolium groups is very nicely illustrated in Figures 1-5. The alternate conformations of the guanidinium, involving 2 or 4 hydrogen bonds, is consistent with the diverse interactions Arg makes in protein structures. Lysine appears to be a special case in that its ammonium group is solvated by one water molecule, which in turn is H-bonded to the cavitand. I suggest a slightly different interpretation of this result, which points to the high charge density on this group compared with the guanidinium and imidazolium cations. Consequently, desolvation of the ammonium group is more unfavourable. Note also the tetrahedral geometry around the ammonium group, completed in this case by the water molecule. The planar guanidinium and imidazolium are less suited to solvation.

Response 1: We thanks the Referee for this suggestion. In the revised manuscript we have added also this possible explanation: “On the other hand, Lys side chain has a higher charge density compared with the Arg and His side chains, in which the conjugation between double bond and nitrogen lone pairs delocalizes the positive charge. Consequently, desolvation of the planar guanidinium and imidazolium groups should be more favourable with respect to the ammonium group of Lys amino acid.”

Point 2: I also found the complexation of the alpha ammonium group interesting. Again, a water molecule solvates the amine. But this time the water is “buried” relative to the plane of the cavitand oxygens. A short comment on this detail would be helpful.

Response 2: We have included this additional observation: “More specifically, in the Lys complex this water molecule is much more inserted in the cavity with respect to the analogous water molecule on the opposing cavitand that hosts the zeta ammonium group (1.15 Å, see Table 1). This different behaviour can be associated with the presence in the α ammonium group complexation a methylene b carbon that covers the cavity (Figure 4a). On the other hand, the complexation of the linear side chain of Lys leaves the water molecule free to adopt a different position in the hydrophobic cavity (Figure 3a).”

Point 3: Considering the dominance of the P=O group in cation binding - is the charge density on the oxygen known? This information might aid in understanding the strength of the hydrogen bonds with the cationic side chains.

Response 3: No theoretical calculation on charge density have been performed.

Point 4: Finally, the introduction could be improved. The opening focus on general biology of Arg, Lys and His is unhelpful. The comment on Tat and Rev is incorrect and some of the cited literature is not the best. It is suggested to focus instead on protein recognition by macrocycles, emphasizing the details of cation bind Response 3: ing. For example, cite also Chem. Commun. 2014, 50, 10412, which treats arginine recognition by sulfonato-calixarene.

Response 4: As commented by the Reviewer, we have removed the comment on Tat and Rev. We have added a sentence on the importance of histidine recognition in proteins, with the corresponding reference (11). Furthermore, we have added another sentence on the use of synthetic macrocycles in protein surface recognition together with three more references, one of which is the Chem Commun suggested by the Reviewer (41-43). All the references were adjusted accordingly. Since the paper is mainly focused on amino acids recognition, we prefer to leave in the introduction few sentences about the biological role of the studied amino acids.

Point 5: It would benefit the community to deposit these structures in the CCDC.

Response 5: The sentence on deposit of these structures has been included.

Point 6: Line 121, replace “In all three structures the chloride anions have been detected at the bottom of the lower rim” with “In all three structures a chloride ion was modelled at the lower rim”.

Response 6: Done

Point 7: Line 161, “deeply inserted” in relation to histidine is an over-statement. Deeply buried might suggest involvement of the phenyl rings of the cavitand.

Response 7: The term “deeply” has been deleted.

Point 8: Define the atom colour scheme once only in Figure 1. It is not necessary to repeat in the other figures.

Response 8: We removed the atom colour scheme from figure 2-5

Point 9: Is Figure 6 really necessary? It appears to be partly a duplication of Figure 1. If retained, use the consistent colour scheme.

Response 9: The Figure 6 has been redrawn accordingly

Point 10: Are references 14-16 necessary? One would suffice.

Response 10: Only Reference 15 has been left

Reviewer 2 Report

The manuscript “Probing the structural determinants of amino acid recognition: X-ray studies of crystalline ditopic host-guest complexes of the positively charged amino acids, Arg, Lys and His with a cavitand molecule" by Brancatelli, Dalcanale, Pinalli and Geremia presents three crystal structures of co-crystals of a phosphonate cavitand as host and three cationic amino acids (Arginine, Lysine and Histidine) as guests. Their main aim was to determine the key interactions manifested in the complexes and to understand the effect of the amino acid side groups in the complex formation.

This aim was achieved and the three crystal structures are well explain. The role of the water molecule in connecting the two hosts in the case of arginine was highlighted. More interestingly was the role of the water molecule interacting with the carboxylic acid of the histidine and resulting in a 4:2 host-gust co-crystal.

The authors have already heavily reported on host-guest chemistry of the phosphonate cavitands with a multitude of different guests including many amino acids. However, the three new structures are interesting and worthy to be reported.

Author Response

We thank the reviewer